# Scoring tissue damage in lung histology images using convolutional neural networks (CNNs)

**Salma Kazemi Rashed and Sonja Aits**

Cell Death, Lysosomes and Artificial Intelligence Group, Department of Experimental Medical Science, Faculty of Medicine, Lund University

sonja.aits@med.lu.se

## Abstract

Histology scoring is a common yet challenging task in both research and healthcare. It still heavily relies on human scorers, which is time-consuming and costly and carries a high potential for errors leading to low reproducibility. Automated scoring with computer vision models could help overcome this problem. Here we demonstrate how CNNs can be used to score lung damage in a porcine model of acute respiratory distress syndrome (ARDS). Models trained by supervised learning performed well, achieving a correlation with ground-truth labels of 0.95 in regression tasks and a weighted F1 score of 0.83 in 3-class classification. In contrast to human scorers, models could also predict damage in smaller regions as opposed to the whole-slide scoring of human scorers, leading to improved resolution of spatial heterogeneity. The major limitation for model performance came from the uncertainty of the ground-truth labels, as human scorers who produced them showed large variability despite being well-trained. This reflects the difficulty of histology scoring and highlights the need for training methods that do not rely on human scorers.

## 1    Dataset

We used Hematoxylin and Eosin (H&E)-stained lung histology images from a porcine model of ARDS (https://www.ebi.ac.uk/biostudies/studies/S-BIAD419) where pigs had been subjected to lipopolysaccharide (LPS)-induced inflammation, in control conditions or in conjunction with mechanical ventilation (MV) and extracorporeal membrane oxygenation (ECMO) treatments, which are typical interventions for human patients with ARDS [1].

Damage scoring had been performed by five biomedical researchers working with lung damage according to the Silva scoring system which considers seven features that indicate lung damage and inflammation: Inflammatory cells, Hyaline membranes, Proteinaceous debris, Thickening of

alveolar wall, Enhanced injury, Haemorrhage, and Atelectasis. Each feature received a score between 0 (no damage) and 8 (highest damage) which were added to obtain a total score.

## 2    Data exploration revealed large variations in human scoring

Lung histology scoring is a very difficult task for human scorers as damage features can be hard to detect and unevenly distributed across the image. When assessing the labels produced by the five human scorers we noticed large variability [1 and unpublished data]. For total scores, correlation between individual scorers ranged from 0.75-0.92. However, for some individual features, the correlations were much lower, with the lowest correlations observed for "Proteinaceous debris" (0.32-0.76) and "Enhanced injury" (0.31-0.72). This highlighted the need for an automated scoring system but was a major limitation for the training of reliable models.

## 3    CNNs perform well in both regression and classification tasks

Models based on the VGG16- and EfficientNet CNN-architectures, which had been pre-trained on ImageNet, were fine-tuned using supervised learning [1 and unpublished data]. Ground-truth labels were generated by averaging the scores given by the five scorers. Images were split into 224x224 tiles with each tile inheriting the label of the parent image. Augmentation was performed by flipping, rotation, mild blurring and brightness variations which mimicked variation occurring in the histology images.

Regression models were first trained using 5-fold cross validation where each of the five treatment groups was used for validation in one of the folds. Not surprisingly, the best results were achieved when the training data included a sample type similar to the one used for validation (e.g. control and mechanical ventilation). The best models were VGG16 models with the simpler top layer (2 fully connected layers + linear layer) and LReLU or ReLU, with a correlation of 0.94 with the ground truth. Next, we trained regression models using 3-fold cross validation where all treatment groups were distributed across both training and validation set but keeping together images from the same animal to avoid information leakage. This resulted in similar performance with the highest correlation with ground-truth labels of 0.95 for an EfficientNetB4 with LReLU. We also trained regression models for predicting the score of individual damage features. Some features were predicted with similar ground-truth correlation as the total score (0.93 and 0.94 for Hemmorhage and Inflammatory Cells) whereas others were slightly more challenging but still resulted in relatively good performance (0.72 and 0.8 for Atelectasis and Thickening of alveolar wall).

Lastly, we trained classification models, by converting the ground truth labels to 3 classes, low, medium and high damage, using a threshold. The best individual model achieved a weighted F1 score of 0.81 which an ensemble of 3 models improved to 0.83. All misclassifications occurred between the medium and the high or low class, whereas no misclassifications were seen between the high and the low class.

To validate predictions and assess the heterogeneity we overlayed prediction results over the full images, with darker patches indicating higher predicted scores (Fig. 1). Predictions showed a large degree of heterogeneity across images. Manual expert review revealed this to correctly reflect

varying degrees of tissue damage across the tissue section. This may have been one of the reasons for the variability in human scoring which only assigned scores on an image level.

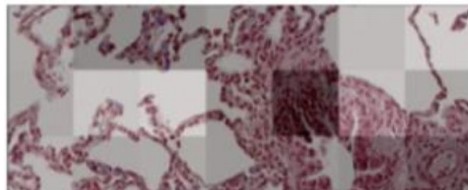

Predicted total image score: 12.5
Ground-truth image score: 12.6

Fig 1. Example image with overlaid tile-based predictions from the best VGG16 model. Darker overlay reflects higher predicted damage score for the tile. Areas with higher damage were identified correctly by the model based on qualitative analysis by human scorers.

As medical image analysis benefits from explainability, we also applied two explainable AI methods, SHAP and Grad-CAM. Damaged regions in the tissue, particularly those with clumps of inflammatory cells, fibroblasts occupying lung airspace, or haemorrhage in the alveolar walls, had the highest SHAP values, indicating their significant impact on the prediction decision for damage, whereas healthy lung tissue regions predominantly had lower SHAP values, reflecting their lesser importance. With Grad-CAM, we observed that in mildly damaged tiles or those where airspace was fully visible, the entire area with alveolar cells or the border of alveolar walls with airspace was highlighted. In contrast, in highly damaged tiles where airspaces are filled with inflammatory cells, the damaged areas (such as clumps of inflammatory cells) were highlighted. In addition, some tiles lacking activation were seen, a reflection of the "dying ReLU" problem. Nevertheless, switching to other activation functions did not significantly improve overall model performance.

## 4    Limitations

Models were limited by the uncertainty of the ground-truth labels which we tried to mitigate by using five scorers for each image to generate an average. In addition, scores were assigned on image level by humans, as tile-based labelling was not feasible at scale, masking the heterogeneity revealed by the tile-based model predictions. Incorrect ground-truth labels were thus likely frequent. In fact, we observed multiple cases where all models agreed but differed from the ground-truth label, indicating a potential mislabeling by human scorers. To overcome these issues, approaches which do not require human scorers should be considered in the future.

## 5    Conclusion

Human lung damage scoring in histology images is a key bottleneck in research as it is time-consuming and costly. It is also a large source of uncertainty as humans struggle with this challenging task even when properly trained, leading to non-reproducible research results. We demonstrated that this task could successfully be taken over by CNNs both for regression and classification tasks. This has the added benefit of revealing heterogeneity in the damage across the samples as models can score smaller regions individually. The findings are likely transferable to many other histology scoring tasks which could benefit from automation.

## Acknowledgement

This study was supported by the Swedish Research Council, the SciLifeLab/Knut and Alice Wallenberg COVID-19 national research program, the Wallenberg AI, Autonomous Systems and Software Program – Humanities and Society (WASP-HS) and Data-driven life science (DDLS) program, the Swedish Research Council for Sustainable Development (FORMAS) and the Crafoord Foundation. The computations were enabled by resources provided by the National Academic Infrastructure for Supercomputing in Sweden (NAISS) and the Swedish National Infrastructure for Computing (SNIC) at Lund University (LUNARC), Chalmers University of Technology (Alvis), and the National Supercomputer Centre at Linköping University (Berzelius, provided by the Knut and Alice Wallenberg foundation), partially funded by the Swedish Research Council through grant agreements no. 2022-06725 and no. 2018-05973.

## Negative societal impact statement

Models trained on limited datasets can propagate biases and make unreliable predictions, which is exacerbated by the somewhat unreliable ground-truth labels. The models are meant as proof-of-principle and should be trained with additional datasets from other sources to improve generalization. There is a risk that these limitations are not considered when models are applied, potentially even on human data.

Furthermore, while automation of histology scoring can improve reproducibility and lower processing times and cost, an overreliance on CNN tools could lead to loss of human skills over time, which will make future model training and evaluation much harder. There is also a negative impact from job losses associated with automation and a risk that the availability of such tools will not be evenly distributed across the world, potentially increasing inequality.

Lastly, model training and deployment consumes resources that can propagate climate change and biodiversity loss even though Sweden itself where the models were trained has a very favourable energy mix.

## References

1. Iran A. N. Silva, Salma Kazemi Rashed, Ludwig Hedlund, August Lidfeldt, Nika Gvazava, John Stegmayr, Valeriia Skoryk, Sonja Aits, Darcy E Wagner. Deep learning for rapid and reproducible histology scoring of lung injury in a porcine model. bioRxiv 2023.05.12.540340; doi: https://doi.org/10.1101/2023.05.12.540340
