# OpenReview forum: "Scoring tissue damage in lung histology images using convolutional neural networks (CNNs)"
_EurIPS.cc/2025/Workshop/MedEurIPS — EurIPS 2025 Workshop MedEurIPS Submission_

### Official Review · Reviewer_s4NU · 2025-10-24
**The work has Limited methodological novelty and is missing interpretability evidence**

**Rating:** 2
**Confidence:** 4

**Review:**

This paper presents an application of convolutional neural networks (CNNs) for automated histology scoring of lung injury in a porcine ARDS model. While the problem is relevant and the motivation is clear, the overall contribution remains quite limited in terms of methodological innovation. The approach largely relies on standard architectures (VGG16, EfficientNet) and supervised regression/classification with averaged human scores as labels, without introducing new modeling strategies, training schemes, or validation concepts beyond what is well-established in medical image analysis.

The manuscript also lacks concrete evidence supporting the interpretability claims in terms of plots of GradCAM or SHAP, which the authors claim were applied.

---

### Official Review · Reviewer_MJ4c · 2025-10-29
**CNNs for Lung Histology  Scoring Highlight Ground-Truth Limitations**

**Rating:** 5
**Confidence:** 4

**Review:**

`Overview:`

This paper applies the *standard VGG16 and EfficientNet models* to automate the scoring of lung tissue damage in Hematoxylin and Eosin histology images from a porcine model of ARDS. The models are trained on labels provided by human experts and demonstrate strong performance in both regression (correlation of 0.95) and 3-class classification (weighted F1 score of 0.83) tasks.

*This work presents a clear application of computer vision (CV) to a foundational task in medical research (histology scoring) that is time-consuming, costly, and subjective.*

- The valuable insight is the paper's analysis of the real bottleneck: the low reliability of the human-generated "ground-truth" labels. *The authors found large variability between human scorers, with correlations as low as 0.31 for some damage features.* This highlights a critical, shared problem in AI in medicine.

- The paper demonstrates a key advantage of the automated approach: the models can provide tile-based scores, revealing spatial heterogeneity of damage that is missed by the "whole-slide" scores assigned by humans.


- The authors themselves correctly identify the main limitation and logical next step: the need for training methods that are robust to, or do not rely on, unreliable human-annotated labels.


`Conclusion:` This is a clear and important contribution. It successfully demonstrates an automated solution while, more importantly, framing the problem in the context of unreliable "gold standards." It is poised to stimulate excellent discussion on a common challenge for the field.

`Suggestions:` Authors can include more details about the model size or the base model source, and a few details about the dataset (size, instance). Future work using an Unsupervised model or a self-supervised CV/ML model.

`Score: 3` Weak reject.

---

### Decision · Program_Chairs · 2025-10-31

**Decision:**

Reject

**Comment:**

Both reviewers agree that the paper addresses a relevant clinical problem and presents solid empirical results using CNNs for histology scoring. However, they find the methodological novelty limited, relying on standard architectures and supervised training.